# Comparison of the Effect of MFAT and MFAT + PRP on Treatment of Hip Osteoarthritis: An Observational, Intention-to-Treat Study at One Year

**DOI:** 10.3390/jcm11041056

**Published:** 2022-02-17

**Authors:** Nima Heidari, Mark Slevin, Yasmin Zeinolabediny, Davide Meloni, Stefano Olgiati, Adrian Wilson, Ali Noorani, Leonard Azamfirei

**Affiliations:** 1The Regenerative Clinic, 18-22 Queen Anne Street, London W1G 8HU, UK; m.a.slevin@mmu.ac.uk (M.S.); yasminzeinolabediny@gmail.com (Y.Z.); adrian@os.clinic (A.W.); alinoorani@gmail.com (A.N.); 2Next AI, London EC3V 1LP, UK; d.meloni@gmail.com (D.M.); s.a.olgiati@gmail.com (S.O.); 3Faculty of Medicine and Pharmacy, The George Emil Palade University of Medicine, Pharmacy, Science and Technology, 540142 Targu Mures, Romania; leonard.azamfirei@umfst.ro; 4Department of Life Sciences, Manchester Metropolitan University, Manchester M15 6BH, UK; 5Department of Morphology, Surgery and Experimental Medicine, Faculty of Medicine, University of Ferrara, 44121 Ferrara, Italy

**Keywords:** micro-fragmented adipose tissue, platelet-rich plasma, osteoarthritis, mesenchymal stem cells, tissue regeneration

## Abstract

Hip osteoarthritis (OA) is a major contributor to reduced quality of life and concomitant disability associated with lost working life months. Intra-articular injection of various biological materials has shown promise in alleviating symptoms and potentially slowing down the degenerative process. Here, we compared the effects of treatment of a cohort of 147 patients suffering from grade 1–4 hip OA; with either micro-fragmented adipose tissue (MFAT), or a combination of MFAT with platelet-rich plasma (PRP). We found significant improvements in both the visual analogue score for pain (VAS) and Oxford hip score (OHS) that were similar for both treatments with over 60% having an improvement in the VAS score of 20 points or more. These results suggest a positive role for intra-articular injection of MFAT + PRP as a treatment for hip osteoarthritis which may be important particularly in low body mass index (BMI) patients where the difficulty in obtaining sufficient MFAT for treatment could be offset by using this combination of biologicals.

## 1. Introduction

Osteoarthritis (OA) is the most common joint disease, affecting more than 250 million people worldwide and being the fourth leading cause of disability in men. Hip OA is a major contributor to the number of years lived with disability worldwide, with the lifetime risk of symptomatic hip OA thought to be around 25 percent [1].

Currently (not taking into account the slow down due to COVID-19), over 95,000 hip replacements are carried out in the United Kingdom each year [2], while worldwide the number is over 1 M. The majority of these are still performed in patients over 65 years of age with around 95% of patients having a normally functioning joint more than 10 years after the operation [3]; however, the prevalence of hip OA is steadily increasing, with studies over the last decade indicating that more than 50 percent of total hip arthroplasties will be performed in patients younger than 65 by 2030 [4].

Hip chondral defects, injuries, and labral tears are the most common sources of hip pain, with femoro-acetabular impingement (FAI) also implicated in the development of hip OA. Recent advances in precision medicine have made it possible to identify earlier stages or pre-symptomatic hip pathology at risk of development of hip OA (although there are no validated diagnostic criteria to date) have provided an infrastructure supporting better management and potentially more treatment options for this pathology [5]. 

Despite the improvement in anatomical design and material composition of prosthesis components, total hip arthroplasty (THA), although highly successful as a surgical intervention, is a highly invasive approach that should be delayed for as long as possible. Hip conservation surgeries, particularly arthroscopic hip surgery, have increased in importance and demonstrated consistent and sustained benefits, however, there is a significant subset of patients (e.g., approximately 22% with FAI) treated in this manner for whom OA symptoms persist, and in addition, treatments such as hip arthroscopy have a similar recovery time to THA surgery [6].

Attempts to alleviate pain and symptoms, thereby delaying THA, beyond the standard protocols involving debridement, labral reconstruction [7] microfracture [8], autologous chondrocyte implantation, matrix-induced chondrocyte implantation, autologous matrix-induced chondro-genesis, and mosaicplasty [9], particularly in younger patients, has encouraged investigation into the potential benefits of cell-based ‘injectables’ in addition to existing cortisone and other non-steroidal anti-inflammatory treatments.

Combined with appropriate physiotherapy, and as an adjunct to traditional surgical procedures, intra-articular injections of hyaluronic acid and platelet-rich plasma (PRP) have been indicated in several studies to be partially effective, at least in relieving pain. Recent meta-analyses have, however, indicated variable outcomes with many clinical trials demonstrating equivalence only when compared with saline placebo [10,11]. Most recently, novel primarily knee OA therapies involving intra-articular injection of either chondrocytes or mesenchymal stem cells (MSCs) through osteo-chondral allograft transplantation or implantation within a synthetic matrix have been used with some success. MSC-based therapies may have the potential to support, repair or even regenerate damaged articular joints [12].

There are many more studies of injectables conducted on KOA than with hip OA and our study here equates, to our knowledge, as the largest study of its type to be recorded [13]. A previous study on 35 patients affected by an acetabular cartilage delamination in femoroacetabular impingement (FAI) treated with micro-fragmented autologous adipose tissue transplantation technique showed improvement in clinical outcomes with a modified Harris Hip Scores significantly higher than microfracture group, over 2 years follow-up.

Micro-fragmented adipose tissue (MFAT) possesses unique biological properties. The adipose tissue has an innate anti-inflammatory quality and contains the highest concentration of MSCs of any tissue in the body (up to 2% of cells sited in the MFAT tissue are MSC compared to a 0.02% in the bone marrow), being derived from the microvessel pericytes, these multipotent cells maintain the capacity to differentiate into chondrocytes with adequate stimuli. 

Recently, there have been several clinical studies where ultrasound-guided intraarticular injection of MFAT have demonstrated significant improvement in pain and mobility in patients with KOA. A recent observational study of 110 OA knees treated with MFAT [14] reporting patient-centered outcomes after 12 months, showed statistically significant improvements in pain, function, and quality of life measured by changes in VAS, OHS, and EQ-5D.

A combination of MFAT with PRP may provide an even more effective therapeutic benefit. In this case, it is believed that the additional anti-inflammatory/pro-angiogenic secretions from the platelets can enhance the overall beneficial effects synergizing with the MFAT delivery payload. Therefore, in this study, we report on the reduction in pain and improvement in function following either a single injection of MFAT or MFAT + PRP for the treatment of hip OA. Our null hypothesis is that there will be no difference in effectiveness between these two treatments.

## 2. Materials and Methods

The study was carried out within a private clinical practice setting. The study was conducted in accordance with the principles of Good Clinical Practice (NIHR) and the General Medical Council (GMC) guidelines on research, patient consent to research, and future publication, in adherence to and in accordance with the Declaration of Helsinki. 

This observational, intention-to-treat study was conducted over a 2-year period and the patients were recruited from 2018 to 2020. The cohort included 57 patients injected with MFAT and 90 patients injected with MFAT + PRP who consented to be scored for pain (visual analogue scale (VAS)) and function (Oxford hip score (OHS)) at baseline irrespective of later changes to adherence or status at follow-up.

Patients had a clinical review and physical examination by an orthopedic surgeon. The Kellgren–Lawrence (KL) system was used to grade hip OA [15] and preoperative assessments included imaging evaluation using X-ray in all patients and MRI in some.

The initial cohort of 57 patients was treated with MFAT alone as this was the normal practice of our private clinic. Following assessment of relevant publications [16,17] demonstrating the potential benefits of combining MFAT and PRP in the treatment of arthritis, it was decided by our clinicians to offer this treatment for the subsequent 90 patients.

### 2.1. Inclusion Criteria

The inclusion criteria were as follows: the presence of hip OA as diagnosed on X-ray and/or MRI.

Exclusion criteria were as follows: recent injury (<3 months) of the symptomatic hip, malignancy, infectious joint disease, anticoagulation, pregnancy or thrombocytopenia, coagulation disorder, as well as intra-articular steroid injections given within the last three months.

The patients were made aware of all other available options for hip OA treatment including conservative interventions, intra-articular injections of steroids, hyaluronic acid, platelet-rich plasma, and MFAT. All patients were made aware of alternative surgical options including hip resurfacing, hip arthroscopy, and THA.

### 2.2. Statistical Methods

A Bayesian analysis was performed for the individual groups of MFAT versus MFAT + PRP indicating variability status of the patient information and appropriate reasoning associated with the group means and variation [18,19].

To our knowledge, there is no existing study comparing MFAT versus MFAT + PRP-specific responses in pain and function to the biologic treatment of hip OA, and on this basis we assumed no prior knowledge on the comparative responses, utilizing minimally informative priors: i.e., normal priors with a large standard deviation for the mean and a broad uniform priors for standard deviation, as described by Kruschke [18,20].

#### 2.2.1. Reproducibility of Analysis and Replicability of Results

In order to make statistical analysis reproducible and results replicable, we utilized open-access software R version 4.0.3 (2020-10-10) and later. In addition, all figures have been generated automatically by software R and are therefore reproducible and replicable.

#### 2.2.2. Missing Values

Our dataset consisted of 57 sets of observations and 8 variables per set of observation for MFAT for a total of 456 data points; and for the MFAT + PRP, 90 patients for a total of 720 data points (Figure 1 and Figure 2). with a missingness rate of 14% (86% observed; 14% missing) for MFAT and 18% (82% observed; 18% missing) for the MFAT + PRP group. These missing values are due to patients being lost to follow-up.

#### 2.2.3. Study Flow Chart

Figure 3 demonstrates the study flow chart detailing the number of patients in each treatment arm, MFAT, and MFAT + PRP.

### 2.3. Patients

The mean patient age at the time of the treatment for the MFAT group was 60 and for MFAT+ PRP was 60 (Table 1). Both groups had a range in grade of hip OA of between 1–4 (median 3) on the KL scale and ASA 1–3 (median of 2; Table 2)). The mean BMI for the MFAT group was 29 and for the MFAT+ PRP was 27. Patients were not separated into different groups for grade of arthritis for the statistical analysis and power calculations due to the overall small numbers of patients in the study. A significant number of patients had severe grade 4 OA at the time of treatment (61/147).

### 2.4. Harvesting Adipose Tissue

In a sedated patient, in an operating theatre, Klein sterile solution (containing saline, Lignocaine, and epinephrine) was injected into the subcutaneous fat. Adipose tissue was then manually harvested, in a standard fashion, via a 13 G blunt cannula connected to a Vaclock 20 mL syringe. The lipoaspirate was injected into and processed using the Lipogems^®^ system. The device is prefilled with saline where stainless steel ball bearings work to mechanically fragment the fat as it is agitated by the clinician. This progressively reduces the size of the clusters of adipose tissue (from spheroidal clusters with a diameter of 1–3.5 mm to clusters of 0.2–0.8 mm). The chamber was then flushed with saline to wash out impurities (e.g., oil, blood, and proinflammatory debris). The resulting product was then filtered through a 500-micron filter making it ready for use [14].

### 2.5. Preparation of PRP

PRP was prepared for each patient using the *Endoret**^®^**(**prgf**^®^**)* Technology (BTI System IV/V; BTI Biotechnology Institute, Vitoria, Spain) [21,22]. Eighteen milliliters of venous blood was taken across two 9 mL tubes containing 3.8% (*w*/*v*) sodium citrate. The presence of 2 tubes allowed for balanced centrifugation. The tubes were centrifuged for 8 min at 580 G (1902 rpm) at room temperature. The 2 mL of plasma located above the buffy coat was collected, with a total PRP volume of 4 mL per patient. The PRP was activated by adding calcium chloride (10% *w*/*v*). This technique has been shown to yield PRP enriched in platelets and reduced in leucocytes.

### 2.6. Injection Protocol

Either 6 mL of MFAT or 4 mL of MFAT + 2 mL of PRP was injected under ultrasonographic guidance into the hip joint. Once the needle was inside the hip joint capsule, it was kept there until the injection of MFAT and PRP was completed. Following full recovery, the patients were discharged with a physiotherapy protocol.

The visual analogue scale (VAS) was used to measure the outcomes for pain and the Oxford hip score (OHS) for function. All patients completed these questionnaires prior to treatment and at three months, six months and one year after the treatment.

The VAS is the most commonly used method allowing individuals to measure and monitor pain intensity using a continuous scale of values. Patients are shown a horizontal line that is anchored by two extremes, between 0 and 100 (0 = no pain, 100 = worst pain). They are then asked to identify the place along the VAS line representing their current level of pain [23].

The OHS has 12 questions, scored 0–4 with 0 classified as severe and 4 as no symptoms, covering pain and function of the hip [24]. Out of the maximum of 48, the highest score means satisfactory joint functions and 0 means severe hip OA [25]. For patients with severe OA, that would have been candidates for arthroplasty.

### 2.7. Responder Classification

The improvement of patients was defined by the following terms: showing an improvement (responder) or no improvement (non-responder) after the inter-articular injection of MFAT ± PRP into the hip. There were three groups in each outcome parameter, those being super-responder, responder, and non-responder. 

For the VAS, individuals without improvement were defined as non-responders, and those improving between 1–19 points higher than pre-treatment on the scale were defined as responders, and those who achieved a score higher than 20 or more were categorized as super-responders [26].

For the OHS, patients who did not improve were designated as non-responders, and those who improved by 1 to 7 points higher than before treatment on the scale were classified as responders. An increase of 8 points or more was classified as super-responder [27].

Super-responder groups consisted of individuals where the degree of improvement in these outcome measures has considered a minimum clinically important difference (MCID).

## 3. Results 

### 3.1. General Outcomes

The data shown here are reported following treatment and analysis from the dataset. The median pre-operative OA grade was 3 for both the MFAT and MFAT + PRP groups, respectively. The mean pre-operative VAS scores were 44 and the mean VAS at 6-months post injection was 28. The full distribution density of these is displayed graphically in Figure 4.

### 3.2. Response to Treatment

#### 3.2.1. VAS

In the MFAT group, of those who completed follow-up, a total of 22 of 35 (63%) responded to the treatment, 14 (64%) being super-responders showing a 20 or more drop in their VAS score for pain (Table 3).

In the MFAT + PRP group, of those who completed follow up, a total of 32 of 44 (73%) responded by showing an improvement to the treatment, with 20 (63 %) being super-responders realizing a 20 or more drop in their VAS score for pain (Table 3).

#### 3.2.2. OHS

In the MFAT group, a total of 22 of 27 (81%) responded to the treatment, with 11 (50%) super-responders showing an improvement of 8 or more in the OHS functional score (Table 3).

In the MFAT + PRP group, 15 of 23 (65%) improved with the treatment, with 11 (73%) super-responders showing an improvement of 8 or more in the OHS functional score (Table 3).

The density distribution of the VAS and OHS was plotted as shown in Figure 4 and Figure 5. The parallel lines show the mean values of VAS and OHS pre (Figure 4) and post (Figure 5) treatment after 1 year. A slight difference in gender performance is noticeable here, with women performing better.

There was a significant reduction in pain in both the MFAT and MFAT + PRP groups. Slightly higher improvements in VAS were recorded in the MFAT group while OHS improved to a similar extent in both groups.

The difference in the means is shown in Table 4 after 1-year treatment. VAS scores showed slightly superior pain reduction and improvement in the MFAT group whilst both MFAT and MFAT + PRP groups showed similar improvements in the OHS.

Conversion to total hip replacement (THR).

In each of the MFAT and MFAT + PRP groups, 10 patients went on to have a THR as they did not respond to the treatment. Of these 20, most had higher grades of OA (KL 3 and 4). Full details are tabulated below (Table 5).

The values shown here demonstrate the entire probability distribution of the difference in the improvement in VAS between the two treatments. The mean of the differences between the treatments is −9.19. The probability of this difference being a meaningful one with the minimal clinically important difference of VAS = > 20 is 12.1%. This difference is not a significant one and suggests that the treatments are equivalent.

The values shown here demonstrate the entire probability distribution of the difference in the improvement in OHS between the two treatments. The mean of the differences between the treatments is −0.145. The probability of this difference being a meaningful one with the minimal clinically important difference of OHS = > 8 is 0.9%. This difference is not a significant one and suggests that the treatments are equivalent.

Figure 6, Figure 7, Figure 8, Figure 9, Figure 10 and Figure 11 show Bayesian plots including the entire uncertainty distribution and variance pre and post MFAT only treatment (1 year); VAS scores with a mean value of −20.8 (Figure 6); OHS at 6.57 (Figure 8).

Similarly, for the MFAT + PRP-treated group, Figure 9 shows a comparable mean OHS change of 7.34; and in VAS of −12.8 (Figure 7); whilst Figure 10 shows the similarity in OHS change between MFAT only and MFAT + PRP with a difference of the means value of 0.145. The values are tabulated in Table 6.

### 3.3. Complications

We did not observe any infections or thromboembolic events. The most common issues included joint pain and pain at the fat harvest site. Joint pain occasionally required more analgesia than was prescribed as a part of the standard postoperative pack.

## 4. Discussion

Here, we have identified similar pain and functional improvements and outcomes between the bioactive substances MFAT vs MFAT + PRP intraarticularly injected into patients with hip OA. 

Our data demonstrated that there were 91 individuals responding with reduced pain in MFAT and MFAT + PRP groups (VAS 63% and 73%, respectively), the number of super-responders was similar between the two groups (VAS 64% in MFAT versus 63% in MFAT + PRP). Pre-treatment, mean VAS scores were 41 for MFAT and 46 MFAT + PRP, indicating a similar level of joint pain, and by the end of this study, the VAS means in both groups improved by a similar amount (15 and 16, respectively). It is also important to note that a total of 20 patients in our cohort went on to have a THA due to the failure of the treatment to improve their condition. These patients had grade 3 and 4 OA. From this small group, we noted that once the hip reaches an advanced state of OA and there is any loss in the sphericity of the joint, the likelihood of improvement with biological therapies diminishes. The hip may indeed be better treated at an earlier stage of arthritis to gain the most benefit.

Hip replacement surgery has been demonstrated to be one of the interventions alongside cataract surgery that brings the most dramatic improvement to the quality of life [28]. Hence, less invasive alternatives such as those described here need to demonstrate high medium to long-term benefits for patients, perhaps underlining the reason why so far, few clinical studies are cited within the literature to date.

In terms of joint function, both MFAT vs MFAT + PRP groups showed notable improvement in OHS at 12 months follow up. Pre-treatment, mean OHS scores were 30 for MFAT and 29 MFAT + PRP, indicating a similar level of joint function, and by the end of this study, the VAS means in both groups improved by a similar amount (6 and 5 points on the scale respectively).

It may be important to note that where PRP was added in the combinationally treated group, concomitantly [16], this meant an equivalent reduction in the amount of MFAT in this mixture compared with the MFAT treatment alone. The suggestion here may be, therefore, that the combination works as effectively as MFAT alone, whilst in general, data do not support such a strong beneficial effect for improvement in patients treated with PRP alone [29]. In our protocol, we used 2 mL of PRP. Further and more detailed analysis of the PRP and MFAT may be helpful in future studies to establish the exact contribution of PRP in this scenario.

From a biological perspective, it is possible that combining PRP with MFAT could synergistically improve the paracrine capacity of the graft. Secretion of complementary cytokines and in some cases identical anti-inflammatory and pro-regenerative molecules such as PDGF and FGF-2 could further promote pain relief whilst enhancing the protective self-response of the joint [30]. In addition, we may hypothesize that platelet granules as well as secreted factors may also contribute to the longer-term drug uptake and releasing capacity of MFAT.

The use of MFAT has already been proven successful in the knee with mild and moderate OA showing improved clinical and functional scores at mid-term follow-up with no treatment-related adverse effects reported. Its ready availability and minimal tissue manipulation allow for maintenance of intact viable MSCs and functional peri-vascular niche in an unaltered micro-architecture, creating delivery of a stable transplantable mini tissue graft [31,32]. 

Regarding the combination of MFAT with PRP, some evidence for the potential therapeutic advantage was provided by Smith et al. [33], who conducted a meta-analysis finding three articles using this combination, successfully treated complex wounds. Similarly, lipo-aspirated fat mixed with PRP was shown to enhance recovery in a case study of a post-menopausal female with lichen sclerosis [34]. 

A recent systematic review of PRP injection alone in patients with hip OA indicated overall improvements in the majority of albeit, limited trials, Ref. [35] whilst most recently, Kraeutler et al. [36] compared Western Ontario and McMaster Universities Osteoarthritis Index (WOMAC) scores and hip flexion from 33 patients treated with either leukocyte-poor PRP (LP-PRP) or low molecular weight HA, finding significant improvements up to 12 months after injection, only in the PRP-treated individuals. From a biological perspective, LP-PRP has the capacity to release notable quantities of anti-inflammatory and pro-reparative factors from the platelet granules and in addition, has been previously shown to stimulate HA production in vitro in patient-derived synovitis [37].

Preliminary data from a study by Dall’Oca et al. [38] examining hips from six patients with hip OA resistant to conservative treatments and with constant pain, showed that MFAT injection significantly reduced WOMAC from 36 to 19 and improved Harris Hip Score (HHS) from 67 to 84 by 6 months with no adverse effects, suggesting the potential inclusion of this technique in future clinical trials.

MSC can act ‘intelligently’ and detect microenvironmental changes acting in a paracrine fashion to release a plethora of growth factors and cytokines with immunomodulatory, antimicrobial, angiogenic, and trophic/regenerative effects on tissue [39,40], encouraging and allowing an optimal, natural dynamic reparative or regenerative response to injury.

Recently, Guo et al. [39] demonstrated prolonged release of large quantities of the anti-inflammatory molecule interleukin-1 receptor-alpha antagonist from MFAT/MSC samples in culture indicating a potential mechanism through which OA-associated inflammation and subsequent pain could be attenuated. A recent systematic review of MSC treatments in dogs with hip OA indicated positive clinical effects including pain reduction and improved mobility and function [41], whilst phase I–II clinical trials in 15 patients with KL grade 2–3 knee OA showed evidence of improved cartilage integrity and regeneration within the joint [42] suggesting a biological interaction that could potentially reverse OA damage.

In this multi-factorial disease, higher body mass index clearly can contribute to additional pressure on joints, and through ethnic, genetic, or medical issues, e.g., diabetes condition increases susceptibility to the development of hip OA at an earlier age [43,44]. 

Compared with knee OA, the investigation of the overall effects of these injectable therapies on hip pain improvement and protection has been neglected. This is one of the first studies highlighting the potential beneficial therapeutic effects of MFA-based treatment in this condition.

Hip OA in women is more often connected to a more complex group of underlying abnormalities and they tend to seek treatment at a later stage; therefore, THA is performed more often in women than in men [45].

Although we have not quantified or included statistical analysis here, overall, women fared better than men and there were more female super responders than male, although the small numbers of individuals within each group and the relatively large missing data points does not allow us to statistically confirm a real-term better response in women. Interestingly, a recent paper by Fossett and Khan [46] showed that stromal and pericyte-derived stem cell numbers were significantly lower in samples taken from women compared with men and this may be one confounding factor in response. 

### Study Limitations

The main limitation of this study is the absence of a control or a PRP alone group in our sample. This self-selected group did not want to have major surgery when they came to our clinic and were treated with an ultrasound-guided single injection of MFAT or MFAT + PRP. 

We included all grades of arthritis. It can be argued that this represents a heterogeneous group of disease. Combining the age range of our cohort (42 to 94) as well as the severity of their conditions (KL grade I–IV) makes for many variables and thus makes subgroup analysis difficult. However, this is a pragmatic representation of our clinical practice, and the highly statistically significant improvement of pain, function, and quality of life cannot be ignored.

The missingness map and study flow diagrams show an attrition rate of 12% in our data collection. Responder fatigue is a well-documented phenomenon and may introduce bias.

In this study, we were unable to perform a gender-bias-mitigated analysis due to missing data points because of poor rate of patient engagement in follow-up.

We did not collect and report compositional data on MFAT and PRP. In future studies, it would be important to include these parameters although many current studies still do not contain this information. 

## 5. Conclusions

In this first of its kind clinical study, we have shown the efficacy of MFAT and combinational preparation in successful amelioration of hip pain together with improved joint function in patients treated with OA. Both MFAT and MFAT + PRP intra-articular injections were equally effective in improving VAS and OHS scores over 6–12 months. A larger clinical trial is warranted in order to characterize in detail the effectiveness in patients with different grades of OA, to determine long-term benefits over 2–5 years, and any gender-related differences in response.

## Figures and Tables

**Figure 1 jcm-11-01056-f001:**
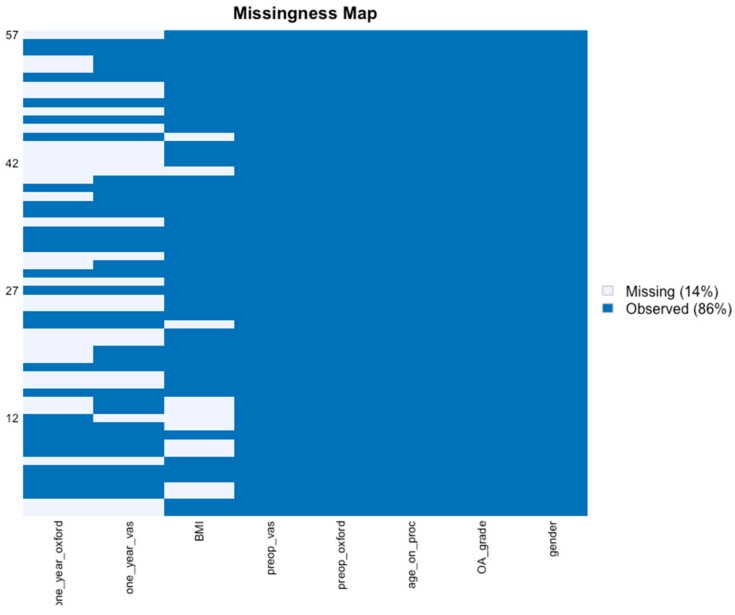
Missingness map MFAT data. Our dataset consisted of 57 sets of observations and 8 variables per set of observation, for a total of 456 data points. Missingness rate of 14% (light blue) due to patient lost-to-follow-up and 86% observed (blue). *x*-axis: outcome variables: Gender, Kellgren-Lawrence Osteoarthritis grade (OA Grade), age at the time of the procedure, Body Mass Index (BMI), Oxford Hip Score (OHS) for function, and Visual Analogue Scale (VAS) for pain at pre-operative baseline and one-year follow-up. *y*-axis: data points missing (14%) due to patients lost-to-follow-up. Source: authors’ data and reproducible statistical analysis with Open Access statistical software R (version 4.0.0 or higher).

**Figure 2 jcm-11-01056-f002:**
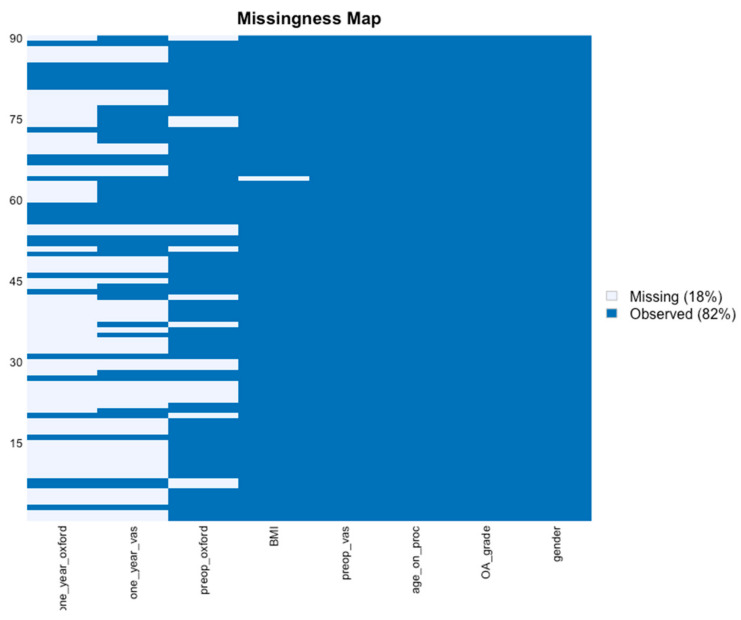
Missingness map MFAT + PRP data. Our dataset consisted of 90 sets of observations and 8 variables per set of observation, for a total of 720 data points. Missingness rate of 18% (light blue) due to patient lost-to-follow-up and 82% observed (blue). *x*-axis: outcome variables: Gender, Kellgren-Lawrence Osteoarthritis grade (OA Grade), age at the time of the procedure, Body Mass Index (BMI), Oxford Hip Score (OHS) for function, and Visual Analogue Scale (VAS) for pain at pre-operative baseline and one-year follow-up. *y*-axis: data points missing (18%) due to patients lost-to-follow-up. Source: authors’ data and reproducible statistical analysis with Open Access statistical software R (version 4.0.0 or higher).

**Figure 3 jcm-11-01056-f003:**
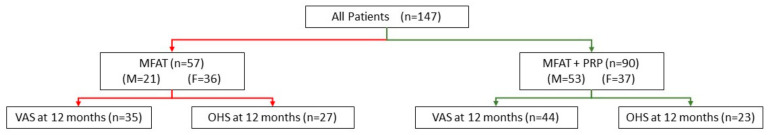
Study flow chart.

**Figure 4 jcm-11-01056-f004:**
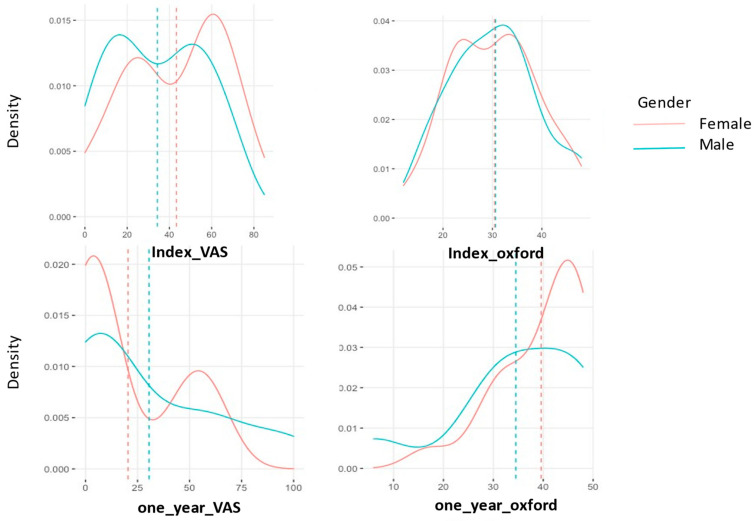
Index (pre-operative) and 1 year (post-operative) VAS and OHS density distribution in patients who received MFAT only. The *x*-axis shows VAS (0–100) and OHS (0–48) pre and 1 years post MFAT injection. The *y*-axis shows the density distribution of the variables.

**Figure 5 jcm-11-01056-f005:**
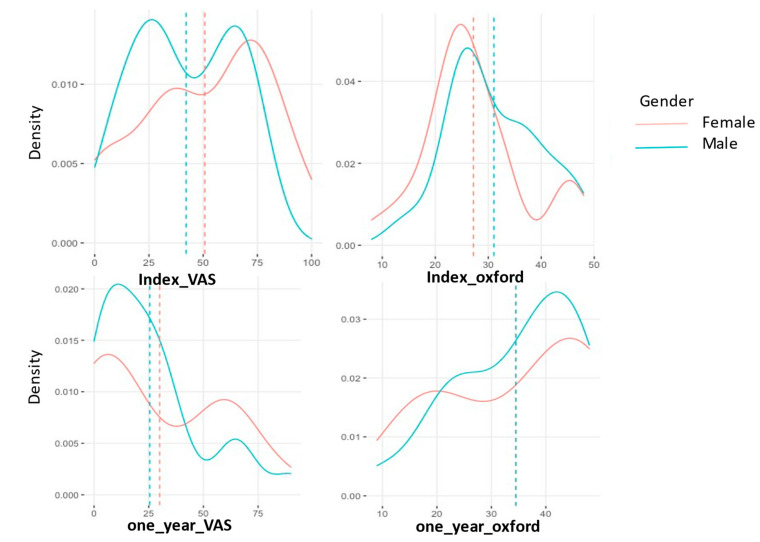
Index (pre-operative) and 1 year (post-operative) VAS and OHS density distribution in patients who received MFAT + PRP. The *x*-axis shows VAS (0–100) and OHS (0–48) pre and 1 years post MFAT injection. The *y*-axis shows the density distribution of the variables.

**Figure 6 jcm-11-01056-f006:**
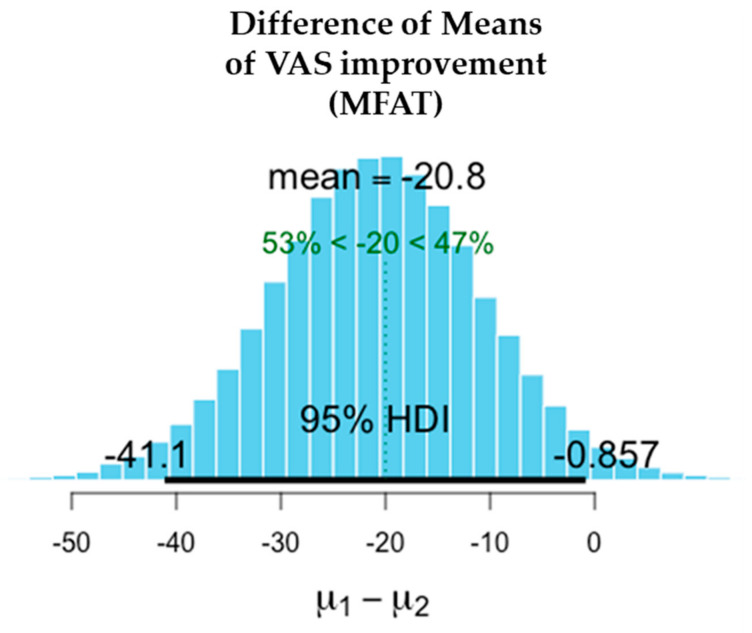
Demonstrates the difference of the means between the pre-treatment (**µ2**) and 1 year post treatment (**µ1**) VAS for MFAT Only. This shows the 95% credible interval between −41.117 and −0.857. The super-responder threshold of −20 is marked the dotted green line.

**Figure 7 jcm-11-01056-f007:**
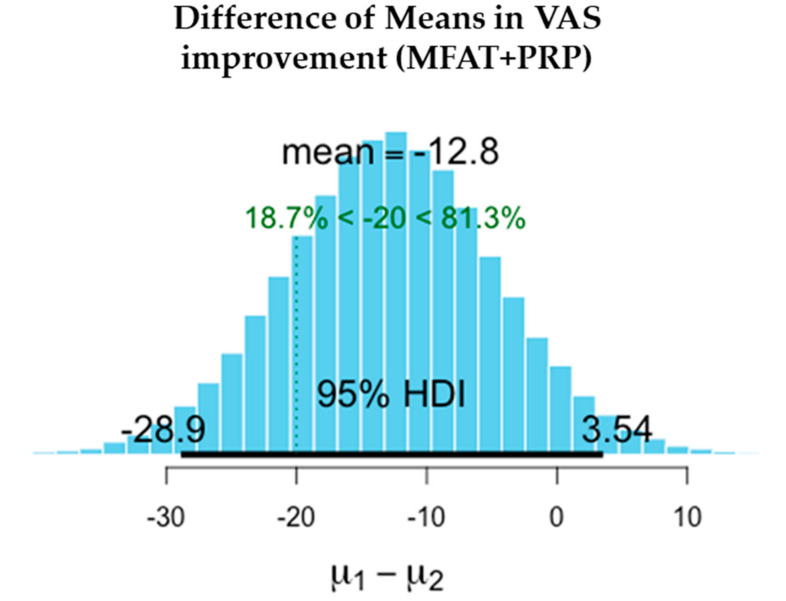
Demonstrates the difference of the means between the pre-treatment (**µ2**) and 1 year post treatment (**µ1**) VAS for MFAT + PRP. This shows the 95% credible interval between −28.882 and +3.543. The super-responder threshold of −20 is marked the dotted green line.

**Figure 8 jcm-11-01056-f008:**
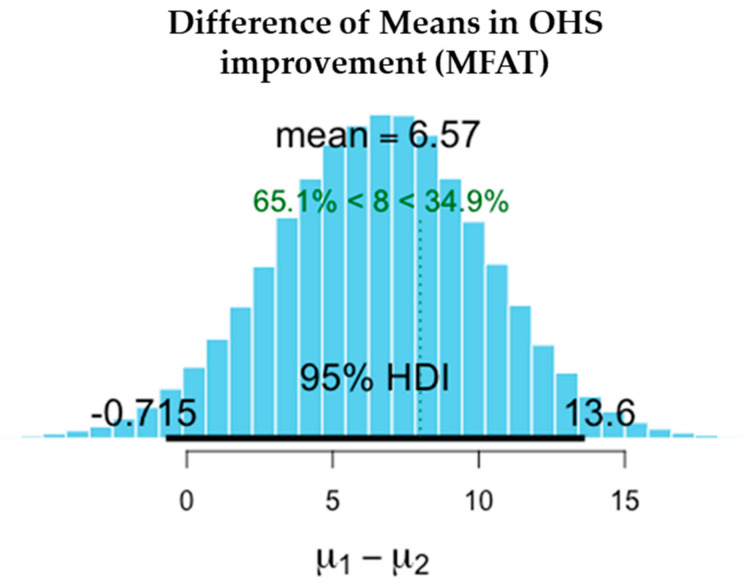
Demonstrates the difference of the means between the pre-treatment (**µ2**) and 1 year post treatment (**µ1**) OHS for MFAT Only. This shows the 95% credible interval between −0.715 and +13.640. The super-responder threshold of 8 is marked the dotted green line.

**Figure 9 jcm-11-01056-f009:**
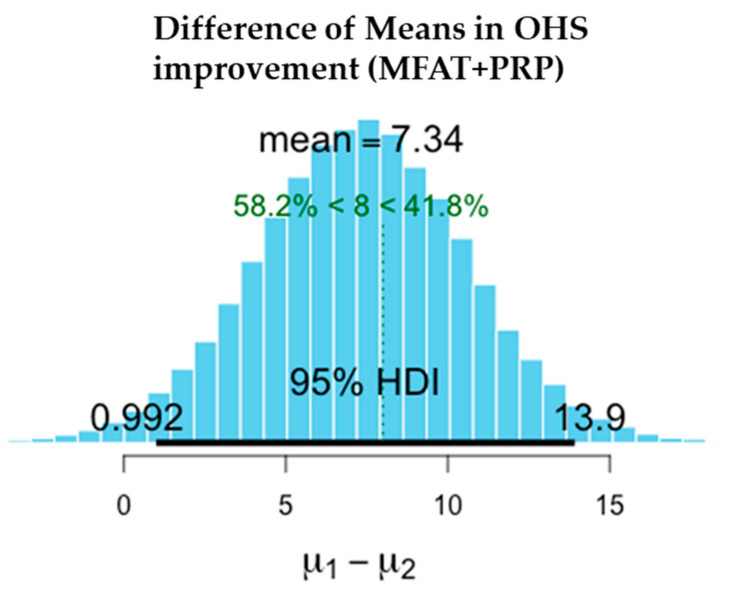
Demonstrates the difference of the means between the pre-treatment (**µ2**) and 1 year post treatment (**µ1**) OHS for MFAT + PRP. This shows the 95% credible interval between +0.993 and +13.920. The super-responder threshold of 8 is marked the dotted green line.

**Figure 10 jcm-11-01056-f010:**
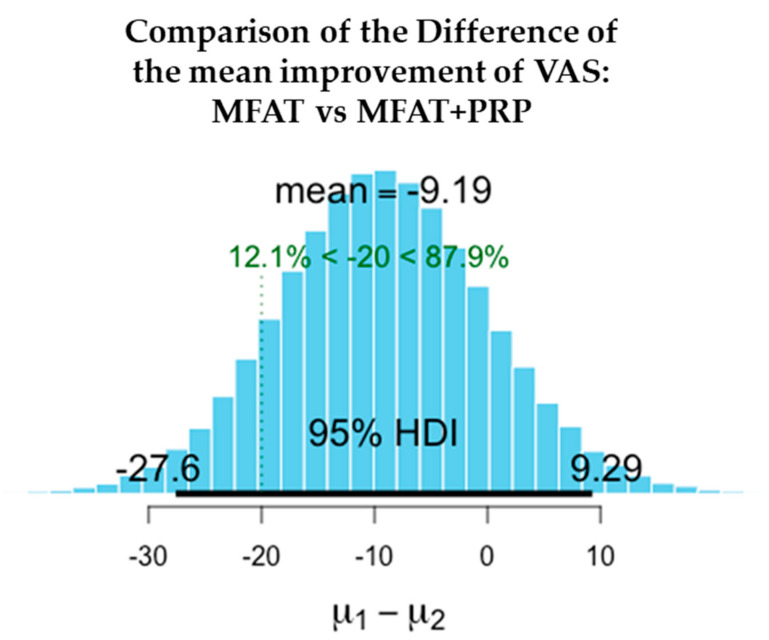
This demonstrates the difference between the change in VAS in MFAT only and MFAT + PRP treatments. (**µ1**): The difference in VAS following treatment by MFAT only (VAS at one year–Index VAS); (**µ2**) the difference in VAS following treatment by MFAT + PRP (VAS at one year–index VAS).

**Figure 11 jcm-11-01056-f011:**
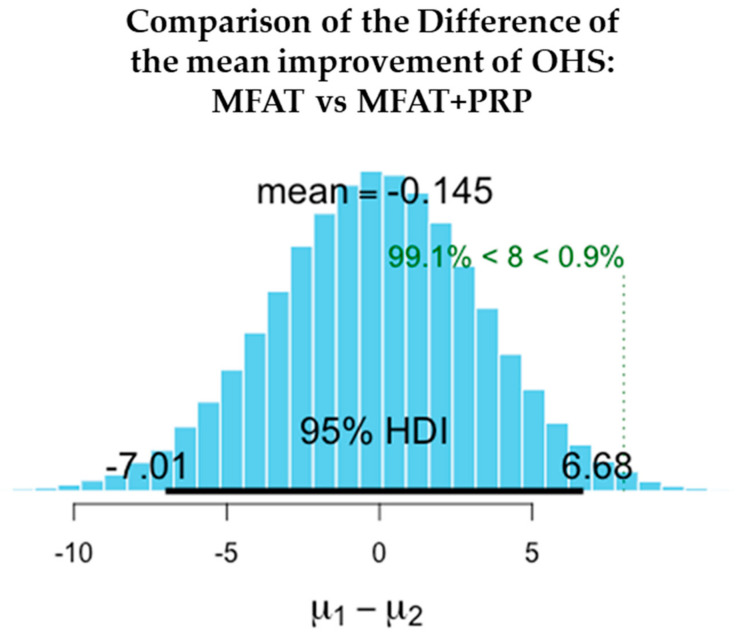
This demonstrates the difference between the change in OHS in MFAT only and MFAT + PRP treatments. (**µ1**) The difference in OHS following treatment by MFAT only (OHS at one year–Index OHS); (**µ2**) the difference in OHS following treatment by MFAT + PRP (OHS at one year–Index OHS).

**Table 1 jcm-11-01056-t001:** Patient demographics. Age and BMI at the time of treatment.

Gender	Treatment	No. of Patients	Mean Age on Procedure (SD)	Mean BMI on Procedure (SD)
FEMALE	MFAT	36	65 (13)	26 (7)
MALE		21	56 (10)	31 (7)
FEMALE	MFAT + PRP	37	60 (8)	26 (4)
MALE		53	60 (11)	27 (5)

**Table 2 jcm-11-01056-t002:** Number of patients in each group according to treatment and grade of OA.

Gender	OA Grade	MFAT	MFAT + PRP
FEMALE	1	6	6
	2	11	7
	3	12	6
	4	7	18
MALE	1	5	8
	2	1	9
	3	5	10
	4	10	26
TOTAL		57	90

**Table 3 jcm-11-01056-t003:** The numbers of super-responders, responders, and non-responders according to criteria for Oxford hip score (OHS) for function, and visual analogue scale (VAS) for pain.

Treatment	Patient Reported Outcome	Super-Responder	Responder	TOTAL Responders	TOTAL Non-Responder	Lost to Follow-Up	Total
MFAT	VAS	14	8	22	13	22	57
	OHS	11	11	22	5	30	
MFAT + PRP	VAS	20	12	32	12	46	90
	OHS	11	4	15	8	67	

Source: Authors’ data and reproducible statistical analysis with open-access statistical software R (version 4.0.0 or higher).

**Table 4 jcm-11-01056-t004:** Improvements in pain and function at 1 year after treatment of hip OA with either MFAT or MFAT + PRP.

Measure	Outcome	Treatment	Difference of the Means	95% Credible Interval of the Difference of the Means	
VAS	Pain reduction (−)	MFAT	−20.771	−41.117	−0.857
		MFAT + PRP	−12.767	−28.882	+3.543
OHS	Function Improvement (+)	MFAT	+6.568	−0.715	+13.640
		MFAT + PRP	+7.339	+0.993	+13.920

**Table 5 jcm-11-01056-t005:** Details of the patients who had a THR following treatment of their hip OA. The Kellgren–Lawrence (KL) grades are shown.

Treatment	Conversion to THR	OA Grade		
		KL 2	KL 3	KL 4
MFAT	10	0	4	6
MFAT + PRP	10	1	2	7

**Table 6 jcm-11-01056-t006:** The difference in the improvements in pain and function at 1 year after treatment of hip OA with MFAT (**µ1**) compared to MFAT + PRP (**µ2**).

Measure	Outcome	Treatment	Difference of the Means	95% Credible Interval of the Difference of the Means	
VAS	Difference in pain reduction (mean)	µ1–µ2	−9.190	−27.610	9.291
OHS	Difference in functional improvement (+)	µ1–µ2	−0.145	−7.011	6.677

## Data Availability

Data are privately held. The authors are happy to consider requests from the editors.

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
