# Peer review of "Comparison of the Effect of MFAT and MFAT + PRP on Treatment of Hip Osteoarthritis: An Observational, Intention-to-Treat Study at One Year"

_jcm, 2022, doi:10.3390/jcm11041056_

Round 1

Reviewer 1 Report

Thanks for the opportunity to review the manuscript entitled "Comparison of the effect of MFAT and MFAT+ PrP on treat-2 ment of Hip Osteoarthritis: An Observational, Intention-to-3 Treat Study at One Year".

The authors compared the clinical benefits of intra-articular injection of micro-fragmented adipose tissues (MFAT) or a combination of MFAT with PRP. The authors found similar improvement in VAS pain score and OHS function score. The manuscript was well designed and written, while some minor issues should be addressed.

  1. As the authors stated in the limitation, the lack of control group raised the concern about the real reason for the positive respond.

  1. The high rate of lost for follow-up also weakened the results and the credibility of the conclusion.

  1. Is there any possibility for the authors to provide more robust evidence for the progression or regenerative effect of hip OA, such as qualitative or quantitative MRI results of HIP.

Author Response

The authors compared the clinical benefits of intra-articular injection of micro-fragmented adipose tissues (MFAT) or a combination of MFAT with PRP. The authors found similar improvement in VAS pain score and OHS function score. The manuscript was well designed and written, while some minor issues should be addressed.

  1. As the authors stated in the limitation, the lack of control group raised the concern about the real reason for the positive respond.

This is a very valid point. It is very well noted that all treatments carry an element of a placebo effect. It is also well known that the placebo effect usually lasts around the maximum of 3 months. Our patient follow-up has been up to 1 year. We therefore surmise that the positive impact of the treatment seen at 1 year is a true outcome from the biological effect of this treatment.

  1. The high rate of lost for follow-up also weakened the results and the credibility of the conclusion.

The high rate of lost to follow-up is unfortunately a feature of real-world data. In my institution we have employed staff to contact patients on a regular basis in order to acquire their outcomes data. Despite this there continue to be a group of patients who do not wish to share their outcomes data with us. We have shown this gap in our data in an honest and open manner as this of course adds an element of bias to the data.

  1. Is there any possibility for the authors to provide more robust evidence for the progression or regenerative effect of hip OA, such as qualitative or quantitative MRI results of HIP.

    Unfortunately it is not possible to acquire any quantitative data with regards to the progression of arthritis in our group. They have now been treated and discharged from our clinic apart from the continued follow-up for research purposes.

Reviewer 2 Report

The present manuscript is a 1-year prospective study comparing efficacy of i/a delivery of micro-fragmented adipose tissue (MFAT), to a combination of MFAT with platelet rich plasma (PrP) in alleviating symptomatic hip OA. It is an agile read and will benefit our readers. Having no control group is a limitation of this study. Nevertheless, research in hip OA is limited and this study will contribute to existing literature. Following are a few comment that I hope the authors take on board.

  1. Citation 21 does not have tissue harvesting technique. The citation should be https://doi.org/10.1155/2020/8881405. An abridged summary of harvest technique will be helpful.
  2. Line 89, 258, 259, 261, 266, 267: OKS should be OHS.
  3. Line 380/388: WOMAC appears earlier than abbreviated 

Author Response

The present manuscript is a 1-year prospective study comparing efficacy of i/a delivery of micro-fragmented adipose tissue (MFAT), to a combination of MFAT with platelet rich plasma (PrP) in alleviating symptomatic hip OA. It is an agile read and will benefit our readers. Having no control group is a limitation of this study. Nevertheless, research in hip OA is limited and this study will contribute to existing literature. Following are a few comment that I hope the authors take on board.

  1. Citation 21 does not have tissue harvesting technique. The citation should be https://doi.org/10.1155/2020/8881405. An abridged summary of harvest technique will be helpful.

    The reference has been altered and an abridged summary has been added to section: 2.4

  2. Line 89, 258, 259, 261, 266, 267: OKS should be OHS.

    This has been amended

  3. Line 380/388: WOMAC appears earlier than abbreviated 

    This has been amended

Round 2

Reviewer 1 Report

None

Academic Editor Notes

The manuscript entitled "Comparison of the effect of MFAT and MFAT+ PrP on treatment of Hip Osteoarthritis: An Observational, Intention-to-3 Treat Study at One Year" deals with the study of the efficacy of novel and increasingly used treatments and is therefore of great interest. The authors have responded satisfactorily to the reviewers' requests. However, they need to resolve the following clarifications before publishing this manuscript:

- The authors should add a CONSORT/STROBE diagram representing the observational study conducted.

This has now been added to the manuscript (Figure 3)

- The authors should explain the criteria on which they based their choice of one type of treatment or another.

Please find an amended paragraph (line 115 – 119):

“The initial cohort of 57 patients were treated with MFAT alone as this was the normal practice of our private clinic. Following assessment of relevant publications [16, 17] demonstrating the potential benefits of combining MFAT and PRP in the treatment of arthritis, it was decided by our clinicians to offer this treatment for the subsequent 90 patients.”

- They should also explain why they included that number of patients (sample size).

The sample size reflects the number of patients with Hip OA treated in our clinic.

- In the results section they have to add data concerning the composition of the applied products (PRP and MFAT). It is advisable to follow the guidelines proposed by the following articles:

Many thanks for the comment. When reviewing the MIBO guidance on reporting within the manuscript we have detailed:

            Patient demographics

            Details on disease diagnosis

            Surgical intervention

            Method of whole blood processing

            Method of PRP processing

            Activation method

Delivery

Post operative care

Outcomes

We however do not have the facility to collect the following information which have not been reported:

            Whole blood characteristics

            PRP characteristics (these have not been stratified routinely measured in any other clinical trial and studies, but in the future, we plan a detailed complex study where all parameters including assessment of individual biological constituent indices will be included)

            Similarly, we have not characterised the MFAT (in the many published articles on this subject including all clinical trials, none have characterized individual patient samples, but this is a complex process which we have done separately in recently published work – and establishing the typical biological properties indicates baseline growth factor and cytokine production in almost all patients’ samples taken sufficient to elicit anti-inflammatory responses in vitro-

Int J Mol Sci

2021 Mar 23;22(6):3271. doi: 10.3390/ijms22063271.

Characterisation of Novel Angiogenic and Potent Anti-Inflammatory Effects of Micro-Fragmented Adipose Tissue. Baoqiang Guo, Xenia Sawkulycz, Nima Heidari, Ralph Rogers, Donghui Liu, Mark Slevin.

PMID: 33806897 PMCID: PMC8004757 DOI: 10.3390/ijms22063271

These are data that we are planning to collect in future clinical trials but our current data does not contain these.

Cells
Murray IR, Geeslin AG, Goudie EB, Petrigliano FA, LaPrade RF. Minimum Information for Studies Evaluating Biologics in Orthopaedics (MIBO): Platelet-Rich Plasma and Mesenchymal Stem Cells. J Bone Joint Surg Am. 2017 May 17;99(10):809-819. doi: 10.2106/JBJS.16.00793

PRP
Kon E, Di Matteo B, Delgado D, Cole BJ, Dorotei A, Dragoo JL, Filardo G, Fortier LA, Giuffrida A, Jo CH, Magalon J, Malanga GA, Mishra A, Nakamura N, Rodeo SA, Sampson S, Sánchez M. Platelet-rich plasma for the treatment of knee osteoarthritis: an expert opinion and proposal for a novel classification and coding system. Expert Opin Biol Ther. 2020 Dec;20(12):1447-1460. doi: 10.1080/14712598.2020.1798925.

- According to the discussion (line 370), it seems that the authors suggest a higher efficacy of cell therapy than PRP. However, this should be tempered as to ensure this there would have to be a group of patients who had received PRP alone (the lack of a PRP group should add to the limitations of the study).

The paragraph that you have referred to reads:

“The values shown here demonstrate the entire probability distribution of the difference in the improvement in OHS between the 2 treatments. The mean of the differences between the treatments is -0.145. The probability of this difference being a meaningful one with the minimal clinically important difference of OHS =>8 is 0.9%. This difference is not a significant one and suggests that the treatments are equivalent.”

Our conclusion is that the 2 treatments are equivalent.

The limitations section has been amended to reflect a lack of a PRP alone group.

- Also according to the protocol, the PRP dose is only 2 mL and a single injection which might be a too low dose to observe a therapeutic effect, so the authors should take this into account in the discussion.

Further detail has been added to the discussion to reflect this point.

- Finally, and in relation to the above, the authors could address in the discussion aspects such as whether PRP could have a positive effect on MFAT cell populations or whether a single injection is sufficient for the application of cellular products, instead of repeated infiltrations.

This is a very important point. We written a paragraph and with an additional reference to address this.

“From a biological perspective, it is possible that combining PRP with MFAT could synergistically improve the paracrine capacity of the graft. Secretion of complimentary cytokines and in some cases identical anti-inflammatory and pro-regenerative molecules such as PDGF and FGF-2 could further promote pain relief whilst enhancing the protective self-response of the joint [30]. In addition, we may hypothesise that platelet granules as well as secreted factors may also contribute to the longer-term drug uptake and releasing capacity of MFAT.”

- Please replace the term PrP with PRP to be in line with the majority of published work.

The manuscript has been amended accordingly